# Equity in wastewater monitoring: Differences in the demographics and social vulnerability of sewered and unsewered populations across North Carolina

Stacie K. Reckling[1,2☯]*, Xindi C. Hu[3☯], Aparna Keshaviah[3]

1 Center for Geospatial Analytics, North Carolina State University, Raleigh, North Carolina, United States of America, 2 Division of Public Health, North Carolina Department of Health and Human Services, Raleigh, North Carolina, United States of America, 3 Mathematica, Inc., Princeton, New Jersey, United States of America

☯ These authors contributed equally to this work.
* skreckli@ncsu.edu

## Abstract

Wastewater monitoring is a valuable public health tool that can track a variety of health markers. The strong correlations between trends in wastewater viral concentrations and county-level COVID-19 case counts point to the ability of wastewater data to represent changes in a community's disease burden. However, studies are lacking on whether the populations sampled through wastewater monitoring represent the characteristics of the broader community and the implications on health equity. We conducted a geospatial analysis to examine the extent to which populations contributing to wastewater collected through the North Carolina Wastewater Monitoring Network as of June 2022 represent the broader countywide and statewide populations. After intersecting sewershed boundary polygons for 38 wastewater treatment plants across 18 counties with census block and tract polygons, we compared the demographics and social vulnerability of (1) people residing in monitored sewersheds with countywide and statewide populations, and (2) sewered residents, regardless of inclusion in wastewater monitoring, with unsewered residents. We flagged as meaningful any differences greater than +/- 5 percentage points or 5 percent (for categorical and continuous variables, respectively) and noted statistically significant differences ($p < 0.05$). We found that residents within monitored sewersheds largely resembled the broader community on most variables analyzed, with only a few exceptions. We also observed that when multiple sewersheds were monitored within a county, their combined service populations resembled the county population, although individual sewershed and county populations sometimes differed. When we contrasted sewered and unsewered populations within a given county, we found that sewered populations were more vulnerable than unsewered populations, suggesting that wastewater monitoring may fill in the data gaps needed to improve health equity. The approach we present here can be used to characterize sewershed populations nationwide to ensure that wastewater monitoring is implemented in a manner that informs equitable public health decision-making.

**Data Availability Statement:** Data used in this study are available from the U.S. Census Data API (https://www.census.gov/data/developers/datasets/census-microdata-api.html) and from the

Agency for Toxic Substances and Disease Registry (https://www.atsdr.cdc.gov/placeandhealth/svi/data_documentation_download.html). North Carolina wastewater treatment plant service areas are available from NC OneMap (https://www.nconemap.gov/datasets/nconemap::type-a-current-public-sewer-systems-2004/about). North Carolina sewershed areas for sites participating in the NC Wastewater Monitoring Network were provided by the NC Department of Health and Human Services (https://www.ncdhhs.gov/).

**Funding:** Financial support for this research was provided by the authors' institutions— Mathematica, North Carolina Department of Health and Human Services, and North Carolina State University. The funders had no role in study design, data collection and analysis, decision to publish, or preparation of the manuscript.

**Competing interests:** The authors have declared that no competing interests exist.

## Introduction

Early in the COVID-19 pandemic, clinical testing was restricted due to mass test kit shortages across the United States. Access to testing—a critical public health resource—was aligned with known structural disparities, with fewer testing sites per person in neighborhoods with more Black, Latinx, and low-income residents, and inequities among minority, uninsured, and rural groups [1, 2]. In poorer areas, testing sites were located farther away [2]. In communities that were majority Black and Hispanic, residents were more likely to face longer wait times and understaffed testing centers, which limited their inclusion in early COVID-19 public health surveillance.

Recognizing that a better way existed to monitor population-wide infection levels, hundreds of communities launched wastewater testing for the SARS-CoV-2 virus that causes COVID-19. Wastewater monitoring can cover a much broader swath of the population than clinical testing, and taps into existing sanitation infrastructure, providing a practical and scalable solution to public health surveillance [3]. In the United States, 16,000 wastewater treatment plants (WWTPs) capture sewage from roughly 75% of the population [4]. Worldwide, researchers estimate that roughly 1 in 4 people is connected to a wastewater treatment plant [5]. Critically, wastewater monitoring captures health biomarkers of sewered populations regardless of whether they visit a testing site or doctor, and regardless of whether they have symptoms since people with asymptomatic infections also shed viral particles into their stools [6, 7].

Despite the potential of wastewater monitoring to improve health equity, resource constraints and a lack of existing wastewater infrastructure may inhibit equitable access to this innovative approach to public health surveillance. Before COVID-19, wastewater monitoring for diseases and controlled substances rarely occurred in low- or middle-income countries (LMICs). Of the fourteen countries that had routinely employed environmental surveillance for poliovirus as part of the Global Polio Eradication Initiative, ten (71%) were high-income countries (HICs) [8]. Likewise, of the 37 member countries in the Sewage analysis CORe group—Europe network, which coordinates international wastewater studies on drug use in and beyond Europe, only 5 (14%) are LMICs [9, 10]. Even after the global expansion of wastewater testing to help officials worldwide manage the coronavirus pandemic, research has shown that monitoring has primarily occurred in HICs [11]. In LMICs, wastewater monitoring is also less likely to be representative of the entire community because sampling is more commonly grab samples collected from surface waters, open drains, or pit latrines (versus composite samples collected from municipal wastewater treatment plants in HICs) [12].

Wastewater monitoring has the potential to overcome disparities in public health surveillance. Indeed, prior research has shown strong correlations between trends in wastewater viral concentrations and trends in COVID-19 case counts countywide, pointing to the ability of wastewater data to represent changes in a community's overall disease burden [13]. However, little research has been conducted to determine the comparability of sewered and unsewered populations with respect to demographics and social vulnerability, and whether communities included in state and national wastewater monitoring programs resemble the larger population. The National Academies Sciences, Engineering, and Medicine report [14], which stressed the importance of equity in national wastewater monitoring efforts, implied that because many unsewered households are in rural areas, and rural areas tend to be more disadvantaged than urban areas, it follows that unsewered populations are more disadvantaged than sewered populations. However, an analysis of data from the 2019 American Household Survey found the opposite to be true—that septic households are more economically advantaged than sewered households, with the pattern upheld even when analyses were stratified by urbanicity [15]. Existing investigations have been hindered by the lack of comprehensive sewershed

geospatial data to define the community areas upstream of wastewater sampling sites. We contribute to the literature by leveraging the sewershed polygon data collected by the North Carolina Wastewater Monitoring Network (NCWMN) to conduct an empirical analysis comparing the sewered and unsewered populations. Given that the Centers for Disease Control and Prevention's National Wastewater Surveillance System (CDC NWSS) will continue to fund state, local, and tribal wastewater programs through at least 2025, characterizing the features of current and future monitored populations can help ensure that wastewater sampling occurs in a manner that promotes health equity.

This study explores the demographic differences between sewered and unsewered populations in North Carolina, one of the first eight states funded by CDC NWSS, and any implications related to the representativeness and equity of wastewater monitoring for public health surveillance. To assess the representativeness of populations contributing to wastewater data collected during the COVID-19 pandemic, we conducted detailed geospatial analyses to answer two key questions: (1) Are sewered populations monitored through wastewater surveillance representative of the counties they come from with respect to demographics and social vulnerability? (2) How similar are the demographics and social vulnerability of communities that are and are not connected to a sewer system (regardless of inclusion in a wastewater monitoring program)? By highlighting the similarities and differences, we aim to improve the use of wastewater data for equitable public health action.

## Methods

To assess the representativeness of populations contributing to wastewater data in North Carolina, we conducted two sets of geospatial analyses. First, we compared the demographic and social vulnerability characteristics of people living in sewersheds (the community area from which wastewater flows to a sampling site) that were monitored by the NCWMN to state- and countywide populations, assessing: (A) monitored sewershed populations aggregated to the state level with the statewide population, (B) monitored sewershed populations aggregated to the county level with their respective countywide population and (C) individual monitored sewershed populations (when multiple wastewater treatment facilities were monitored within a county), with their respective countywide population. Second, we compared the demographics and social vulnerability of sewered versus unsewered populations within a given county to evaluate the comparability of populations that can and cannot contribute to wastewater monitoring. In total, we analyzed the sewershed population of 38 WWTPs in 18 counties, including 25 actively monitored WWTPs as of June 2022, one previously monitored WWTP, and 12 WWTPs not monitored by the NCWMN (S1 Fig).

### Data collection and pre-processing

The NCMWN monitored sewershed boundary polygons were obtained from the North Carolina Department of Health and Human Services. For the analysis of sewered versus unsewered populations, we delineated sewered and unsewered polygons for nine counties for which we could wholly identify the county's sewershed boundary geospatial data for all municipal WWTPs with a treatment capacity of more than 0.5 million gallons per day. To create a single county-level sewered area polygon, we merged NCWMN-monitored sewershed polygons with additional sewershed polygons for WWTPs not monitored by the NCWMN which were available from NC OneMap [16]. We then created unsewered county area polygons by erasing the sewered polygon from the county polygon. The nine counties covered rural and non-rural counties from across the state and included eight counties actively participating in NCWMN as of June 2022 plus Chatham County, which had previously participated in NCWMN.

To summarize population demographics and social vulnerability for sewered and unsewered populations, we grouped 23 variables to represent five conceptual domains: demographics, health, housing and transportation, social vulnerability indices, and socio-economic status (SES) (S1 Table). Most variables clearly belonged to one of the five domains, while others straddled multiple domains. We grouped English proficiency within SES because language skills are often related to educational attainment and job prospects. Variables describing race and ethnicity came from the 2020 United States Census Redistricting Data, which were available at the block level geography. We also analyzed variables from the 2015–2019 American Community Survey (2015–19 ACS) that captured age, gender, health insurance status, level of education, wealth, English proficiency, housing, employment, and disability status, all of which were available at the tract level geography. To prepare the data for geospatial analysis, we cleaned and joined tabular Census data to TIGER/Line tract or block polygons [17]. Lastly, we downloaded a shapefile of Census tracts with information on the CDC's social vulnerability index (SVI) including the overall SVI percentile rank and the ranks for each of the four SVI themes (socioeconomic status, household composition and disability, minority status and language, and housing type and transportation). We filtered the data to include only the counties in the study area described above.

## Geospatial analysis

In a geographic information system (GIS) we assessed the demographics, SES, and SVI of populations residing in the various geographies of interest: individual sewersheds, sewersheds aggregated by county, sewersheds aggregated by state, counties, the state, sewered county areas, and unsewered county areas. To do this, we selected census blocks or tracts that intersected each polygon of interest using a spatial intersect. While dissolving the selected tracts or blocks into a single polygon based on a common attribute (in this case state), we summed variables representing population counts and averaged variables representing population percentages. Then, we calculated summary statistics, including percentages that showed the share of the total population represented by different demographic groups, the average median household income, and population-weighted averages of SVI ranks. All analyses were performed using either Arc-GIS Pro 2.9 [18] or R version 4.1.3 [19] using the *sf* [20] and *tidycensus* [21] packages.

We compared the characteristics of different groups by calculating the percentage point (pp) difference for categorical variables and the percent (%) difference for continuous variables (median household income). We designated potentially meaningful differences between populations using a threshold of +/- 5 pp for categorical variables and +/- 5% for continuous variables. We chose this approach to be conservative and ensure that we did not overlook smaller disparities that were within the reported margin of errors (MOEs). This is especially relevant for a health-equity-focused analysis because smaller groups often have larger MOEs, but a lack of statistical significance should not be interpreted as a lack of meaningful findings. For completeness, we assessed statistical significance by comparing differences to twice the reported MOEs, for 2015–19 ACS variables. The 2020 Census data and SVI data did not include MOEs at the time of this analysis.

## Results

### Characteristics of sewersheds participating in the NC Wastewater Monitoring Network

Sewersheds for WWTPs participating in the NCWMN as of June 2022 covered a broad geographic area of the state and had populations that ranged from 3,500 to 550,000 people.

**Table 1. North Carolina sewersheds monitored by the NCWMN as of June 2022.**

| Sewershed name | County name | Sewershed population | County population | % of the county population monitored |
|---|---|---|---|---|
| Laurinburg | Scotland | 15,527 | 34,823 | 45% |
| Tuckaseigee | Jackson [a] | 13,296 | 43,109 | 31% |
| Marion | McDowell | 8,459 | 45,756 | 18% |
| Beaufort | Carteret [a] | 3,500 | 69,473 | 5% |
| Roanoke Rapids | Halifax | 14,320 | 69,493 | 21% |
| City of Wilson | Wilson | 49,384 | 81,801 | 60% |
| Chapel Hill–Carrboro | Orange | 78,141 | 148,476 | 53% |
| Greenville | Pitt [a] | 89,616 | 180,742 | 50% |
| Wilmington City | New Hanover [a] | 58,361 | 234,473 | 25% |
| New Hanover County (North) | New Hanover [a] | 67,743 | 234,473 | 29% |
| South Durham | Durham [a] | 108,105 | 321,488 | 34% |
| Fayetteville -Rockfish Creek | Cumberland | 151,589 | 335,509 | 45% |
| MSD of Buncombe County | Buncombe | 173,000 | 378,608 | 46% |
| Winston Salem—Salem | Forsyth [a] | 178,000 | 382,295 | 47% |
| Jacksonville | Onslow | 41,819 | 204,576 | 20% |
| Greensboro, North Buffalo | Guilford | 135,821 | 537,174 | 25% |
| Charlotte 1 | Mecklenburg [a] | 68,685 | 1,110,356 | 6% |
| Charlotte 2 | Mecklenburg [a] | 182,501 | 1,110,356 | 16% |
| Charlotte 3 | Mecklenburg [a] | 120,000 | 1,110,356 | 11% |
| Raleigh | Wake [a] | 550,000 | 1,111,761 | 49% |
| Raleigh 2 | Wake [a] | 37,020 | 1,111,761 | 3% |
| Raleigh 3 | Wake [a] | 7,648 | 1,111,761 | 1% |
| Cary 1 | Wake [a] | 84,189 | 1,111,761 | 8% |
| Cary 2 | Wake [a] | 74,331 | 1,111,761 | 7% |
| Cary 3 | Wake [a] | 75,886 | 1,111,761 | 7% |

Note: Sites are listed in order of ascending county population size. MSD = Metropolitan Sewerage District.

[a] Indicates counties included in the sewered vs unsewered county analysis. Chatham County was not being monitored as of June 2022 so it is not included here, but it is included in the sewered versus unsewered analysis.

Monitored sewershed populations accounted for 1% (Raleigh 3) to 60% (City of Wilson) of a county's population and 31% of the state's population (Table 1). In three of the 17 counties analyzed, multiple sewersheds were monitored, which together accounted for 33% (Mecklenburg), 54% (New Hanover), and 75% (Wake) of the respective county's population. More detailed environmental metadata of the wastewater monitoring program can be found in S2 Table and a previous publication [13].

## Populations in monitored sewersheds versus state- and countywide

As a whole, the populations residing in the 25 sewersheds monitored through NCWMN as of June 2022 resembled the statewide population. For the following 15 of 23 variables we analyzed, differences amounted to less than +/- 5 pp or 5%: demographics (percent female, percent African American, percent Asian, percent American Indian/Alaska Native, percent Native Hawaiian or Pacific Islander, percent 65 years and older, percent Hispanic), health status (percent with disability, percent without health insurance), housing and transportation (percent of households without a vehicle, percent group quarters), social vulnerability index (housing and transportation vulnerability), and SES (percent below federal poverty line, percent unemployed, percent limited English proficiency) (Fig 1A). However, populations in

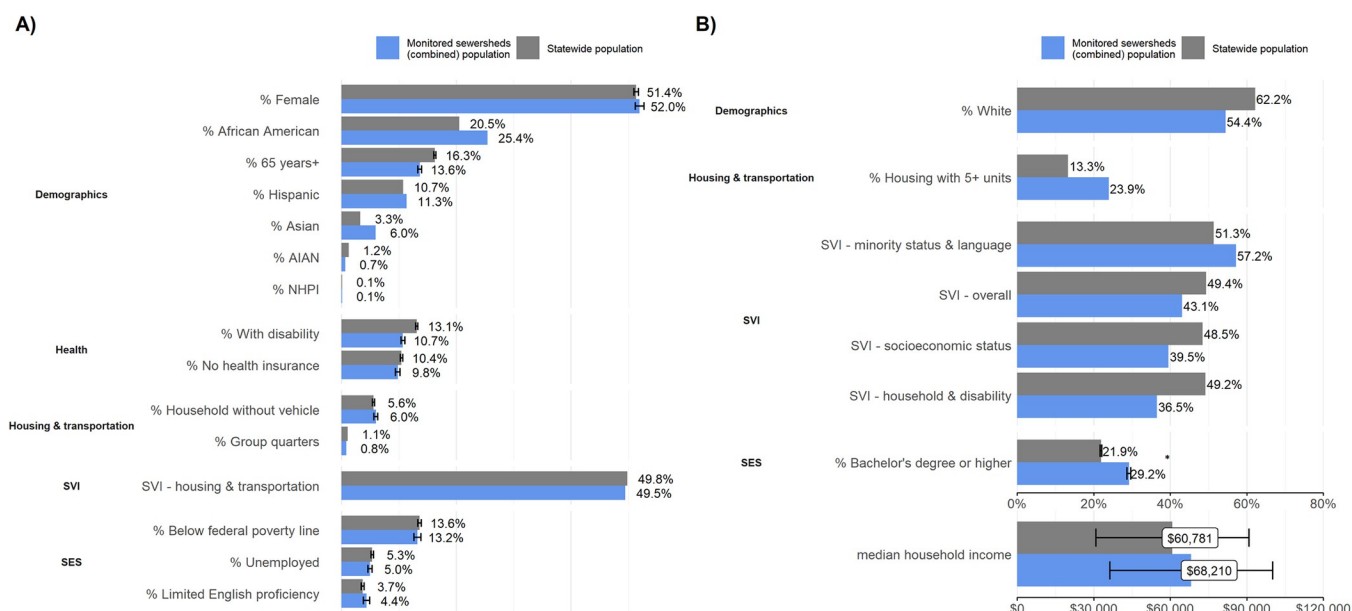

**Fig 1. Demographic differences between monitored sewersheds and the statewide population.** Plots show A) variables with less than a 5% or pp difference and B) variables with greater than a 5% or pp difference. The error bars represent the 95% confidence interval which was only calculated for variables in the ACS 2015–2019 data because the MOE information wasn't available for the 2020 census at the time of the analysis. Variables with a statistically significant difference are indicated by an asterisk (*).

monitored sewersheds across the state had fewer White residents (i.e., more minorities), lower social vulnerabilities (overall, SES and household composition and disability), more housing with five or more units, greater educational attainment, and higher median household income compared to the statewide population (Fig 1B). These differences were only moderately outside the +/- 5 pp or % threshold (ranging from -6.0 to +13.0 pp or %) and only educational attainment reached statistical significance (S3 Table). The observed differences may be related to how sites were enrolled in North Carolina's wastewater monitoring program. The initial group of sites participating in the NCWMN came from a COVID-19 wastewater surveillance pilot project coordinated by universities [22], and so were located in urban centers near universities with labs that had the capacity to analyze wastewater samples. Over time, the NCWMN expanded to include sites in other areas of the state, including the rural mountainous region in Western North Carolina, and underserved communities with higher social vulnerability, low COVID-19 vaccination rates, or both [23].

When we compared populations living in monitored sewersheds, after aggregating within the county, to their respective countywide populations, we found that monitored sewershed populations generally resembled their countywide population. Differences were not meaningfully different for the following 11 of 23 variables analyzed: demographics (percent female, percent over 65 years old, percent Asian, percent Native Hawaiian or Pacific Islander, percent Hispanic), health status (percent without health insurance, percent disability), housing and transportation (percent of households without a vehicle), and SES (percent limited English speaking, percent below federal poverty, percent unemployed). There was a meaningful difference between at least one combined monitored sewershed and the county for the remaining 12 variables analyzed, with the largest differences relating to race, social vulnerability, median household income, and housing with greater than five units (Fig 2). Monitored sewershed populations had a lower share of White residents compared to countywide populations in 12 of 17 counties (with meaningful differences in three), while African Americans made up a

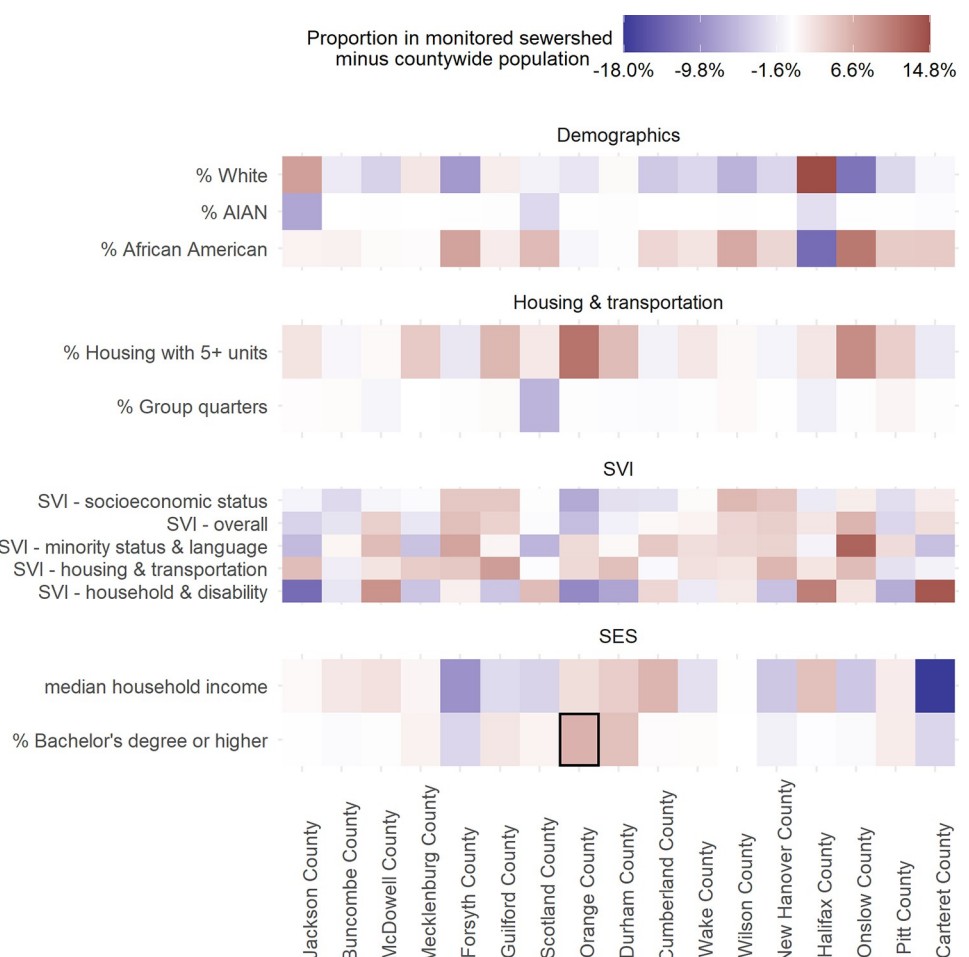

**Fig 2. Demographic differences between monitored sewersheds and the respective countywide population.** Only demographic variables with more than a +/-5% or pp difference (monitored–county) are included. Counties are displayed from west to east based on the location of the county centroids. Shades of red indicate the variable is higher in the monitored sewershed population while shades of blue indicate the variable is higher in the county population. Blocks highlighted with a black outline are both meaningfully different and statistically significantly different.

higher share of the monitored sewershed population in 14 of 17 counties (with meaningful differences in four). The greatest differences in race generally occurred in sewersheds in the eastern part of the state. However, in Jackson County, located in western NC, we also observed a meaningful difference in race, where a lower share of American Indian and Alaska Natives resided in the sewershed compared to the county (note: Jackson County borders the Qualla Boundary, which is home to the sovereign nation of the Eastern Band of the Cherokee Indians). Overall SVI ranks were similar between monitored sewershed and countywide populations, but minority and language vulnerability and housing and transportation vulnerability were higher in the majority of monitored sewersheds (Fig 3). The difference in median household income ranged from -19.8% to +5.8% where nine sewershed populations had higher median household incomes and eight sewershed populations had lower median household incomes. Housing with five or more units was higher in 11 monitored sewersheds (four were meaningfully different).

In the three counties in which multiple sewersheds were monitored, we noted differing degrees of similarity between individual sewershed populations and the countywide

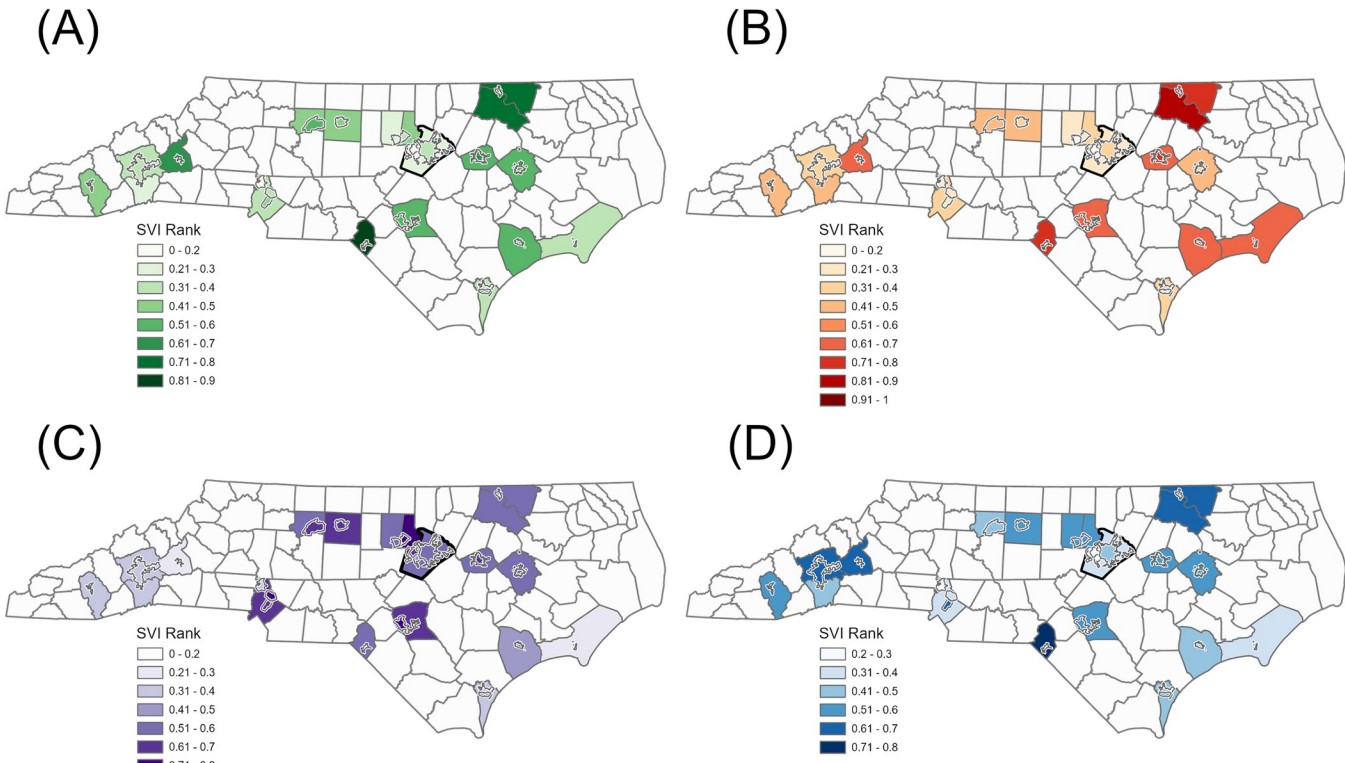

**Fig 3. Social vulnerability of populations in individual monitored sewersheds versus countywide.** Maps show individual monitored sewershed and county population SVI ranks for the four SVI themes: (a) socioeconomic status, (b) household composition and disability, (c) minority status and language, and (d) housing type and transportation. Wake County is shown with a bold outline. North Carolina county boundaries can be downloaded from https://www. nconemap.gov/.

population. In all three counties, we observed meaningful differences in race, median household income, social vulnerability, educational attainment, and housing with five or more units that were not evident when the individual sewersheds were combined and analyzed as a single geographic unit (S5 Table). In Wake County, the combined sewershed SVI ranks resembled the county SVI ranks even though the six individual sewersheds showed a wide range of SVI ranks across all four themes: socioeconomic status (individual ranged from 0.12–0.51, combined = 0.27, county = 0.27), household composition and disability (individual = 0.16–0.74, combined = 0.28, county = 0.29), minority status and language (individual = 0.44–0.76, combined = 0.59, county = 0.56), and housing and transportation (individual = 0.25–0.63, combined = 0.42, county = 0.40) (Fig 3). Notably, residents in two Wake County sewersheds, Raleigh and Raleigh 3, appeared to be more disadvantaged than other Wake sewersheds and countywide residents, given their higher social vulnerability overall and across all themes, coupled with lower educational achievement and lower median household income.

## Sewered versus unsewered populations

In a second set of analyses, we compared the characteristics of sewered and unsewered populations in nine counties for which we could obtain complete sewershed boundary geospatial data. We found that 19 variables meaningfully differed in at least one county (Fig 4), and only four variables (percent Asian, percent Native Hawaiian and Pacific Islander, percent female, and percent unemployed) did not meaningfully differ. Most notably, we found differences in racial and ethnic makeup, median household income, and social vulnerability. In most of the

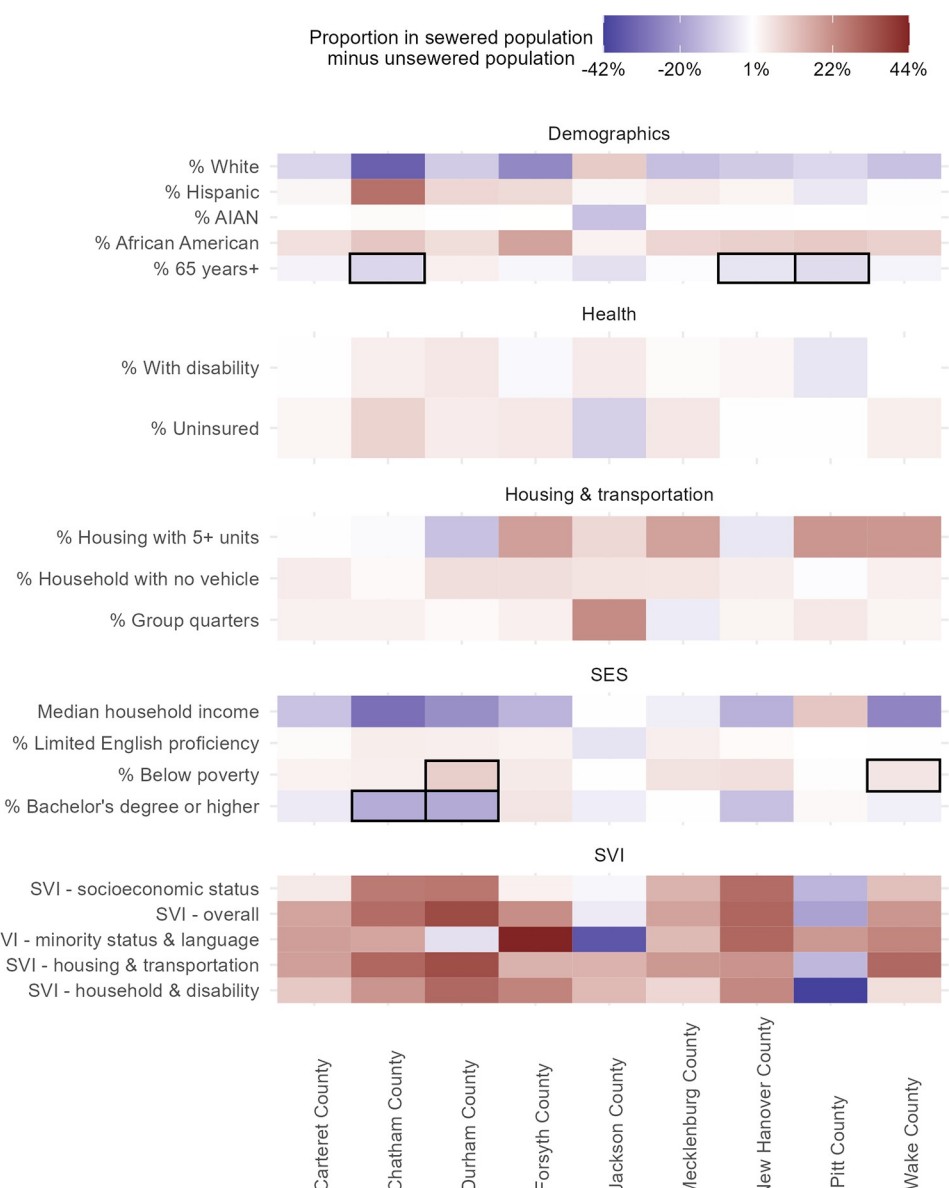

**Fig 4. Demographic differences between sewered and unsewered county residents.** Only demographic variables with more than a +/-5% or pp difference (sewered–unsewered) are shown. Counties are displayed from west to east based on the location of the county centroids. Shades of red indicate the variable is higher in the sewered population while shades of blue indicate the variable is higher in the unsewered population. Blocks highlighted with a black outline are both meaningfully different and statistically significantly different.

nine counties, Hispanics and African Americans made up a greater share of the sewered population than the unsewered population with up to a 28.2 pp difference in the share of Hispanics and up to an 18.3 pp difference in the share of African Americans. Conversely, sewered populations had a lower share of White residents. We also found that in all but one county, the median household income was lower in the sewered population than the unsewered population, with differences ranging from -30.0% to -0.2%. Educational attainment was lower in the sewered population than the unsewered population in seven counties (all but Forsyth County and Pitt County), ranging from a -7.0 to -0.2 pp difference. This difference was also statistically

significant for Durham and Chatham counties (S4 Table). Finally, in seven of nine counties (all but Jackson County and Pitt County), we found that overall social vulnerability and vulnerability based on each of the four SVI themes were higher among the sewered population than the unsewered population.

## Discussion

Wastewater data can be used to track disease trends in communities connected to a sewer system. However, research describing how sewered and unsewered populations differ is limited. One study utilized WWTP sewershed polygon data to examine differences in population sizes, but did not look at social vulnerability or demographic differences [24]. Another explored demographic and economic characteristics of US households connected to sewer aggregated to various Census geographies, but did not utilize sewershed boundary data [25]. The present study is the first to combine sewershed polygon data and several population-based spatial datasets to characterize sewered and unsewered populations and assess whether populations monitored by a state's wastewater program represent broader populations. Our findings suggest that residents of sewersheds monitored by the NCWMN as of June 2022 represent broader North Carolina populations well. Comparisons between populations in monitored sewersheds and state- or countywide residents, showed that many of the variables analyzed (11 of 23) did not differ meaningfully, and when differences were found, they generally occurred in only a few counties and were only slightly above the +/- 5% or pp threshold. The level of similarity we found, which extended across all domains (demographics, health, housing and transportation, SES, and social vulnerability indices), indicates that wastewater data collected through the NCWMN at the time of this study accurately represented state and county populations. Further, the strong correlation between county level and sewershed level COVID-19 clinical data points to the reliability of wastewater monitoring as a public health tool [13, 26]. While studies on sewershed population fluctuations are still needed, a simulation study suggests that wastewater data from sewered communities can be indicative of health trends in neighboring unsewered areas when cases are widespread [25].

In a few instances, we found meaningful differences between sewered residents and unsewered or countywide residents, which has implications related to health equity. First, our analyses highlighted that minority populations may be over-represented in the state's wastewater data. African Americans represented a higher share of monitored sewershed residents than countywide, while Whites often comprised a smaller share of monitored sewershed residents than county or statewide. Also, vulnerability related to minority status and language was greater in monitored sewersheds than statewide. These results suggest that, by better representing potentially vulnerable racial and ethnic minorities, wastewater data may have filled critical gaps in clinical case data during the pandemic. Early in the pandemic, case data underrepresented Black and Hispanic communities, and even in the summer of 2022 (when the Omicron variant was dominant), the severe undercounting of COVID-19 cases was more pronounced among Black and Hispanic populations, as well as among younger adults ages 18 to 24 and those with lower income and less education [27]. Despite the potential benefits from an equity lens, having a higher share of minority residents in monitored sewersheds versus county or statewide populations creates a risk of inaccurate health messaging to the public. Because racial and ethnic minorities have seen higher SARS-CoV-2 infection rates than White, non-Hispanic populations [28, 29], wastewater data that overrepresents these groups could lead to inflated COVID-19 infection estimates. Another notable finding was that educational attainment was significantly higher in the monitored sewershed population versus statewide (and countywide in one county) but was meaningfully lower in sewered versus unsewered residents

within a county. The implications of these findings are important to consider because lower educational attainment is associated with lower receptivity to public health messaging and higher vaccine hesitancy [30]. Finally, we found that North Carolina's sewered populations had greater overall SVI, a higher share living below the poverty level, and significantly lower educational attainment, compared to unsewered residents. In other words, sewered populations in the nine counties analyzed may be more at-risk than unsewered populations, suggesting that if all municipal wastewater systems within the county were monitored, the resulting wastewater data would be more likely to capture the health information of vulnerable populations. Whether the disproportionate representation of vulnerable populations in wastewater monitoring is desirable depends on how the data will be used, but the potential overrepresentation is important to recognize when interpreting and communicating insights from wastewater data. Furthermore, given these disparities, wastewater monitoring data should be interpreted alongside other surveillance data to gain a more complete picture of the 'true' state of public health.

Our analysis was subject to several limitations. The geospatial methodology may have misclassified some residents as belonging to the sewershed population. This is because, to aggregate data to the sewershed level, we utilized a spatial intersect which selected census tracts or blocks that touched the sewershed polygon boundary. When tracts or blocks partially extended outside the sewershed boundary, this method may have overestimated the count of persons in the sewershed. Future studies could use hi-resolution gridded population data, when available, to more accurately determine populations in the sewershed [31]. Also, because statewide septic system location data are not readily available, we assumed that all homes inside the sewershed boundary were connected to the sewer even though some might utilize onsite septic systems. Likewise, it is important to interpret our findings in the context of known limitations and biases in the underlying US Census data. Data on race and ethnicity collected during the 2020 US Census were subjected to a new disclosure avoidance system called differential privacy, which added an unknown amount of statistical noise to the published data products to shield sensitive information from discovery [32]. We aggregated Census block data to larger geographies, which should minimize inaccuracies associated with differential privacy. Moreover, the Demographic Analysis, one of the leading indicators of data quality for decennial censuses, showed a record undercount of Hispanics during the 2020 Census [33]. Although we did not discover a meaningful difference in the share of the Hispanic population between monitored sewersheds and the county or the state, it is possible that undercounting obscured any potential difference. Finally, our comparison of North Carolina's sewered and unsewered residents was limited to nine counties. A comprehensive North Carolina sewershed dataset would enable us to confirm that sewered populations tend to be more vulnerable than unsewered populations throughout the state.

In our analysis, we assumed that all people residing within monitored sewersheds contributed to the wastewater data collected by the NCWMN. However, some people connected to monitored sewered systems could be excluded from wastewater data. People who shed little or no virus in their feces will not be represented in wastewater data, and preliminary research suggests that demographic and geographic features may influence viral shedding rates. For example, early in the pandemic, Parasa et al. [34] found that fecal shedding rates varied substantially across eight studies included in their meta-analysis, estimating that, on average, 41% of confirmed COVID-19 cases (range = 17% to 80%) shed the virus in their stools. More recently, Prasek et al. [35] noted differences in estimated shedding rates across communities of differing ages, ethnicity, and socioeconomic composition, as well as over time, as the dominant variant changed (though it is worth noting that these findings were subject to ecological fallacy and lacked the use of multivariate regression modeling to control for confounding factors).

Also, communities that utilize on-site wastewater management, such as septic systems, will be missing from wastewater monitoring data even if programs expand to include other WWTPs. This may be of particular importance in states like NC where roughly 50% of state residents use septic systems [36].

## Future perspectives

As wastewater monitoring expands in geographic reach and utility, it will become increasingly important to describe in detail the populations residing in sewered areas. Given previous research showing that the relationship between income and the use of decentralized wastewater systems varies across states [37], state wastewater programs should evaluate the characteristics of sewered and unsewered populations and carefully consider the implications of any differences to ensure that wastewater monitoring is executed equitably within the state. We recommend that wastewater programs periodically reassess the representativeness of the monitored population as the number of sites changes or new US Census data are released. A change in site composition is especially significant for counties with multiple monitored sewersheds because we observed that characteristics of residents in individual monitored sewersheds often differed from the county. Removing sampling sites or adding new sampling sites in these counties could impact the degree of similarity between the monitored sewershed and county populations.

The geospatial methods described in this study could be readily adapted to other states if a national sewershed database were developed, perhaps by building on the updated Clean Watershed Needs Survey conducted by the EPA [38]. This study's methodology could also be expanded to include additional US Census variables or other indices, such as the area deprivation index [39], social deprivation index [40], and structural racism effect index [41], which are relevant to public health in that state or region. Recognizing that sewershed populations are dynamic, wastewater monitoring programs should factor in known fluctuations in the size of the sewershed population due to seasonal tourism or major events, and broader changes in demographics and population mobility [25] when interpreting changing wastewater trends. Although we found that areas with greater sewer connectivity have lower income, research has shown that in some states, the opposite is true and sewer connectivity decreases with decreasing income [42]. Accordingly, when expanding wastewater monitoring to new sampling locations, officials should consider the role of structural inequalities and environmental justice [42]. One potential approach to enhancing the equity of wastewater monitoring for public health would be to consider whether sampling occurs in areas of high COVID-19 disease burden. When disease hot spots and sewer connectivity do not overlap geographically, a sampling approach combining monitoring at centralized treatment plants with sampling at sentinel locations (such as schools and offices) [43] could improve the representativeness of wastewater data.

A better understanding of the characteristics of populations included in wastewater monitoring will also help officials use wastewater data effectively and adapt sampling strategies to address ongoing public health needs. Insights into the demographics and vulnerability of wastewater-monitored populations can enable tailored communications and interventions and equitable resource distribution. Furthermore, wastewater programs can use population information to effectively monitor additional pathogens such as influenza, respiratory syncytial virus, and monkeypox virus. Although broad, representative sampling is desirable when monitoring for pathogens like respiratory illnesses, which spread throughout the population each year, sampling specific sewershed populations may be more suitable for other health markers. For example, wastewater monitoring programs seeking to fill gaps in traditional

public health surveillance data may focus on including vulnerable sewershed populations that lack healthcare resources, or monitoring sewersheds with high tourism rates where clinical data doesn't reflect true disease prevalence. Likewise, to enable early outbreak control, it might be most useful to monitor select sewersheds where populations are at risk or cases have previously concentrated.

## Conclusion

Evidence-based public health decisions need to be informed by complete, high-quality data that equitably represent the community. Our analyses confirmed that wastewater data collected across North Carolina represents county and state populations well. Further, wastewater monitoring has the potential to improve health equity by better capturing the health information of vulnerable populations compared to clinical data. The in-depth geospatial analyses described here provide a framework for evaluating the characteristics and representativeness of wastewater-monitored populations and can be adapted as additional geospatial data describing sewered areas and population characteristics become available. Understanding sewered population characteristics will help officials use wastewater data effectively for public health decision-making and adapt wastewater testing strategies to monitor for future pathogens.

## Supporting information

**S1 Fig. North Carolina sewershed map.** A sewershed boundary shows the area from which wastewater flows to a wastewater treatment plant sampling site. Monitored sewersheds were those participating in the North Carolina Wastewater Monitoring Network as of June 2022. Monitored sewersheds were combined with unmonitored sewersheds to create a single sewered area polygon for the county. North Carolina county boundaries are found at https://www.nconemap.gov/.
(TIF)

**S1 Table. Descriptions of analyzed variables.**
(XLSX)

**S2 Table. Meta-information on wastewater infrastructure, sampling, and testing methods.** The table shows meta-variables that substantially differed across sites. Wastewater sample analysis was conducted by the following three labs: University of Wisconsin-Milwaukee (Tuckaseigee), North Carolina State University (Raleigh 2, Raleigh 3, Cary 1, Cary 2, and Cary 3), and University of North Carolina-Chapel Hill (remaining sites). Sampling generally occurred twice weekly, though was often less frequent around holidays, and occurred only weekly in Tuckaseigee before August 2021. The concentration method used was membrane filtration with $MgCl_2$ (all sites) and acidification (all sites except Tuckaseigee). The extraction method used the NUCLISENSE manual magnetic bead extraction kit (all sites except Tuckaseigee) or bead bashed HA filters on a KingFisher Flex system 96 well plates (Tuckaseigee only). All sites shared the following features: Sample location type = wastewater treatment plant, System type = separated, Sample mix = raw wastewater, pre-concentration storage temp = 4°C, PCR type = Digital droplet polymerase chain reaction (ddPCR), SARS-CoV-2 targets = N1 and N2, recovery control name = Bovine coronavirus (BCoV) vaccine, endogenous control = Pepper mild mottle virus (PMMoV), extraction blanks = yes. For additional details on sample analysis methods, see previous publications (1, 2). Sites are ordered by ascending county population size. Note: Flow = 24-hr flow-weighted composite; MGD = Million gallons per day; MSD = Metropolitan Sewerage District; Time = 24-hr time-

weighted composite.
(XLSX)

**S3 Table. Differences in characteristics between the monitored sewershed and the county-wide or statewide populations.** Rows with bolded fonts mean the difference is statistically significant at a significance level of 0.05. Rows shaded in grey mean the difference between the sewered and unsewered population is meaningful (greater than 5 percentage points or 5%). Note: The margin of error (MOE) for the monitored and whole populations was calculated using the MOE estimates published by the Census Bureau and following the formula: square root of the sum of squared margin of errors. Statistical significance is assessed based on the absolute value of the Z statistics (*** for $|Z| > 3.29$, ** for $|Z| > 2.58$, * for $|Z| > 1.96$).
(XLSX)

**S4 Table. Differences in characteristics between the sewered and unsewered populations.** Rows with bolded fonts mean the difference is statistically significant at a significance level of 0.05. Rows shaded in grey mean the difference between the sewered and unsewered population is meaningful (greater than 5 percentage points or 5%). Note: The margin of error (MOE) for the monitored and whole populations was calculated using the MOE estimates published by the Census Bureau and following the formula: square root of the sum of squared margin of errors. Statistical significance is assessed based on the absolute value of the Z statistics (*** for $|Z| > 3.29$, ** for $|Z| > 2.58$, * for $|Z| > 1.96$).
(XLSX)

**S5 Table. Characteristics with meaningful differences between populations in individual monitored sewersheds and the county population.** Values are shown for individual monitored sewersheds as well as monitored sewersheds aggregated within the county for counties where multiple sewersheds participated in the NC Wastewater Monitoring Network. Only variables with more than a +/-5 percent (%) or percentage point (pp) difference (monitored sewershed—county) in at least one individual monitored sewershed vs county comparison are included. Variables that have a statistically significant difference are indicated by an asterisk (*).
(XLSX)

## Acknowledgments

The authors would like to acknowledge the contributions of several partners to this work, including Virginia Guidry, Ariel Christensen, and Steven Berkowitz from the North Carolina Department of Health and Human Services.

## Author Contributions

**Conceptualization:** Stacie K. Reckling, Xindi C. Hu, Aparna Keshaviah.

**Data curation:** Stacie K. Reckling, Xindi C. Hu.

**Formal analysis:** Stacie K. Reckling, Xindi C. Hu.

**Investigation:** Xindi C. Hu.

**Methodology:** Stacie K. Reckling, Xindi C. Hu, Aparna Keshaviah.

**Validation:** Stacie K. Reckling, Xindi C. Hu.

**Visualization:** Stacie K. Reckling, Xindi C. Hu.

**Writing – original draft:** Stacie K. Reckling, Xindi C. Hu, Aparna Keshaviah.

**Writing – review & editing:** Stacie K. Reckling, Xindi C. Hu, Aparna Keshaviah.

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
