## [Decision Letter · Decision Letter 0]

28 May 2024

PONE-D-24-16609Equity in wastewater monitoring: Differences in the demographics and social vulnerability of sewered and unsewered populations across North CarolinaPLOS ONE

Dear Dr. Reckling,

Thank you for submitting your manuscript to PLOS ONE. After careful consideration, we feel that it has merit but does not fully meet PLOS ONE’s publication criteria as it currently stands. Therefore, we invite you to submit a revised version of the manuscript that addresses the points raised during the review process.

We look forward to receiving your revised manuscript.

Kind regards,

Renjith VishnuRadhan, PhD

Academic Editor

PLOS ONE

Journal Requirements:

"Financial support for this research was provided by the authors’ institutions—Mathematica, North Carolina Department of Health and Human Services, and North Carolina State University. The funders had no role in study design, data collection and analysis, decision to publish, or preparation of the manuscript."

3. We note that Figures 3 and S1 in your submission contain map images which may be copyrighted. All PLOS content is published under the Creative Commons Attribution License (CC BY 4.0), which means that the manuscript, images, and Supporting Information files will be freely available online, and any third party is permitted to access, download, copy, distribute, and use these materials in any way, even commercially, with proper attribution. For these reasons, we cannot publish previously copyrighted maps or satellite images created using proprietary data, such as Google software (Google Maps, Street View, and Earth). For more information, see our copyright guidelines: http://journals.plos.org/plosone/s/licenses-and-copyright.

a. You may seek permission from the original copyright holder of Figures 3 and S1 to publish the content specifically under the CC BY 4.0 license.  

Additional Editor Comments:

Dear Dr. Stacie Reckling,

Thank you for submitting your manuscript to PLOS ONE. 

Reviewers' comments on your paper entitled "Equity in wastewater monitoring: Differences in the demographics and social vulnerability of sewered and unsewered populations across North Carolina" have now been received. You will see that they are advising revision of the manuscript. I suggest to consider these comments in your revised version of manuscript.

If you decide to revise the work, please submit a list of changes or a rebuttal against each point which is being raised when you submit the revised manuscript. After manuscript resubmission, it will be reviewed again.

Best regards,

Handling Editor

Reviewers' comments:

Reviewer's Responses to Questions

**Comments to the Author**

1. Is the manuscript technically sound, and do the data support the conclusions?

Reviewer #1: Yes

Reviewer #2: Yes

2. Has the statistical analysis been performed appropriately and rigorously? 

Reviewer #1: Yes

Reviewer #2: Yes

3. Have the authors made all data underlying the findings in their manuscript fully available?

Reviewer #1: Yes

Reviewer #2: Yes

4. Is the manuscript presented in an intelligible fashion and written in standard English?

Reviewer #1: Yes

Reviewer #2: Yes

5. Review Comments to the Author

Reviewer #1: The authors have presented quite a comprehensive, descriptive and informative set of information on the manuscript entitled ‘Equity in wastewater monitoring: Differences in the demographics and social vulnerability of sewered and unsewered populations across North Carolina’.

The authors may address the following queries:-

Introduction section can be revised and include a discussion addressing the following points-

• What were the main challenges to accessing clinical testing during the early COVID-19 pandemic, and how did these challenges align with existing structural disparities?

• What were the main challenges to accessing clinical testing during the early COVID-19 pandemic, and how did these challenges align with existing structural disparities?

• What are the potential limitations and equity concerns associated with wastewater monitoring, particularly in low- and middle-income countries (LMICs)?

Methodology Section:-

• In what ways could the methodology used in this study be improved or expanded to provide a more comprehensive understanding of the representativeness and equity of wastewater monitoring across different regions?

In the results and discussion section, the following points can be explained with relevant citations:-

• How reliable is wastewater monitoring as a public health tool in representing the health markers and disease burden of an entire community, especially given the variations in sewershed populations?

• What are the implications of finding that sewered populations are more vulnerable than unsewered populations on the overall effectiveness of wastewater monitoring in reflecting the true state of public health?

• How do the demographic and social vulnerability differences between sewered and unsewered populations affect the accuracy and equity of health data derived from wastewater monitoring?

• What measures can be taken to ensure that wastewater monitoring includes a representative sample of the entire population, especially in areas with significant differences between sewered and unsewered residents?

A section of future perspective can be included in the manuscript and following points can be highlighted:-

What are the potential limitations of relying on wastewater monitoring for public health surveillance, and how can these limitations be mitigated to improve the accuracy and inclusivity of the data?

How might the differences between individual sewershed and county populations impact the interpretation of wastewater monitoring data for public health decision-making at the county and state levels?

How can the findings of this study inform the development of policies and practices to enhance the use of wastewater monitoring as a tool for equitable public health surveillance and intervention?

What role do socio-economic and infrastructural factors play in the observed differences between sewered and unsewered populations, and how can these factors be accounted for in future wastewater monitoring studies?

Reviewer #2: This manuscript looks at differences in the demographics of sewered and unsewered populations across North Carolina. From the wastewater-surveillance and wastewater-based epidemiology aspect, I'm really unsure what the purpose is because of course the samples only represent those who contributed to them. There are other publications that have used census data and wastewater analysis to identify correlates of demographics which may have otherwise remained unknown which is what I was hoping this study would contribute to - but these seem to have been overlooked. As such I don't really see how this adds valuable information to the literature - but it is something that obviously should be considered when interpreting data in. It is written well though and the methodology used to determine catchment demographics is correct. BUT it also ignores that catchments are dynamic and that changes occur both in terms of short term and longer term - which is also important consideration for monitoring.

6. PLOS authors have the option to publish the peer review history of their article (what does this mean?). If published, this will include your full peer review and any attached files.

Reviewer #1: **Yes: **Devlina Das Pramanik

Reviewer #2: No

---

## [Author Response · Author response to Decision Letter 0]

8 Jul 2024

Thank you for giving us the opportunity to submit a revised draft of the manuscript “Equity in wastewater monitoring: Differences in the demographics and social vulnerability of sewered and unsewered populations across North Carolina” for publication in the journal PLOS One. We addressed the editor's and reviewers' comments point-by-point in a response to reviewers letter uploaded as a separate document.

---

## [Decision Letter · Decision Letter 1]

6 Aug 2024

PONE-D-24-16609R1Equity in wastewater monitoring: Differences in the demographics and social vulnerability of sewered and unsewered populations across North CarolinaPLOS ONE

Dear Dr. Reckling,

Thank you for submitting your manuscript to PLOS ONE. After careful consideration, we feel that it has merit but does not fully meet PLOS ONE’s publication criteria as it currently stands. Therefore, we invite you to submit a revised version of the manuscript that addresses the points raised during the review process.

We look forward to receiving your revised manuscript.

Kind regards,

Renjith VishnuRadhan, PhD

Academic Editor

PLOS ONE

Journal Requirements:

Reviewers' comments:

Reviewer's Responses to Questions

**Comments to the Author**

1. If the authors have adequately addressed your comments raised in a previous round of review and you feel that this manuscript is now acceptable for publication, you may indicate that here to bypass the “Comments to the Author” section, enter your conflict of interest statement in the “Confidential to Editor” section, and submit your "Accept" recommendation.

Reviewer #1: All comments have been addressed

Reviewer #3: (No Response)

2. Is the manuscript technically sound, and do the data support the conclusions?

Reviewer #1: Yes

Reviewer #3: Yes

3. Has the statistical analysis been performed appropriately and rigorously? 

Reviewer #1: Yes

Reviewer #3: I Don't Know

4. Have the authors made all data underlying the findings in their manuscript fully available?

Reviewer #1: Yes

Reviewer #3: Yes

5. Is the manuscript presented in an intelligible fashion and written in standard English?

Reviewer #1: Yes

Reviewer #3: Yes

6. Review Comments to the Author

Reviewer #1: (No Response)

Reviewer #3: Equity in wastewater monitoring: Differences in the demographics and social vulnerability of sewered and unsewered populations across North Carolina

The authors have developed a framework that can be used for assessing the demographic and social vulnerability of sewered and unsewered populations for diseases with the help of wastewater monitoring data. Through geospatial analysis, the authors have shown that the wastewater monitoring data can be representative at the state or county level using demographic details from the US census. The authors have appropriately responded to the Editors and reviewers’ comments during the first revision and adequately revised the manuscript. The work carried out by the authors is unique and can be considered for publication with minor revisions. I have some questions or suggestions to improve the quality of the manuscript.

1. One of the outcome of the study is that the sewered population is vulnerable than unsewered population. Is that because for the unsewered population, water quality is not monitored? Is it monitored? How is it monitored? Can that add to the data gap?

2. In the case of COVID-19, even at the household level, if not all the members were infected, then how can wastewater monitoring be helpful than clinical assessment, which would provide individual-level assessment to take precautions? Although wastewater monitoring would be helpful in identifying county or state-level severity of the spread of disease; at the local or individual level, clinical assessment would be important. Then what is the contribution of this study? For whom the information obtained through this assessment would be beneficial?

3. The results from the study can be briefly justified based on the authors’ experience or understanding

a. For example-

Line 260-263 - Educational attainment was lower in the sewered population than the unsewered population in seven counties (all but Forsyth County and Pitt County), ranging from a -7.0 to -0.2 pp difference. This difference was also statistically significant for Durham and Chatham counties (Table S4).

What is the reason for this? How is this expression related to the aim of the study? The authors can explain the results after stating them with the reason as per their experience or understanding.

4. The discussion section explains more about the result, limitations and future work; however, a generalized discussion about integrating the findings of the study to provide a comprehensive picture and its importance in the context of existing literature needs to be added apart from just one the statement in line number 319-320 “Furthermore, given these disparities, wastewater monitoring data should be interpreted alongside other surveillance data to gain a more complete picture of the ‘true’ state of public health”

5. Conclusion, is a stand-alone section that should also include the limitation and future scope. In the conclusion section, the authors have precisely stated the outcomes of the study. However, the authors also need to summarize the limitations and future scope to make it a distinct section for effective readership.

6. Minor corrections:

a. At some places % or pp is written, be consistent about its use throughout the manuscript

b. Supplementary sheet – Include the title, authors and affiliation at the top of the supplementary sheet so that the file doesn't get lost when a reader downloads it.

7. PLOS authors have the option to publish the peer review history of their article (what does this mean?). If published, this will include your full peer review and any attached files.

Reviewer #1: **Yes: **Devlina Das Pramanik

Reviewer #3: No

---

## [Author Response · Author response to Decision Letter 1]

18 Sep 2024

Dear Editor and Reviewers, 

Thank you for allowing us to submit a revised draft of the manuscript “Equity in wastewater monitoring: Differences in the demographics and social vulnerability of sewered and unsewered populations across North Carolina” for publication in the journal PLOS One. Please see below where we have addressed the comments point-by-point and made changes to the manuscript.

Editor’s Comments to the Author:

Point 1: Journal Requirements:

Author response: Thank you for bringing this to our attention. We checked that our references are published and accessible via the DOIs and URLs. We updated the following references’ URLs:

Olesen SW, Young C, Duvallet C. The Effect of Septic Systems on Wastewater-Based Epidemiology. 2022 Oct. Available: https://biobot.io/publications/the-effect-of-septic-systems-on-wastewater-based-epidemiology/

CDC. Cases, Data, and Surveillance. In: Centers for Disease Control and Prevention [Internet]. 2023 [cited 26 Apr 2023]. Available: https://archive.cdc.gov/#/details?url=https://www.cdc.gov/coronavirus/2019-ncov/covid-data/investigations-discovery/hospitalization-death-by-race-ethnicity.html

Reviewer’s Comments to the Authors:

Reviewer #1: (No Response)

We are glad we have addressed Reviewer #1’s previous comments. We want to thank them for the helpful feedback.

Reviewer #3:

• Point 1: One of the outcome of the study is that the sewered population is vulnerable than unsewered population. Is that because for the unsewered population, water quality is not monitored? Is it monitored? How is it monitored? Can that add to the data gap?

Author response: We appreciate this comment but would like to clarify that our paper focuses on comparing the demographics and social vulnerability of sewered and unsewered populations. Water quality was not included in the scope of this study. 

• Point 2: In the case of COVID-19, even at the household level, if not all the members were infected, then how can wastewater monitoring be helpful than clinical assessment, which would provide individual-level assessment to take precautions? Although wastewater monitoring would be helpful in identifying county or state-level severity of the spread of disease; at the local or individual level, clinical assessment would be important. Then what is the contribution of this study? For whom the information obtained through this assessment would be beneficial?

Author response: Thank you for your suggestion. We agree with the reviewer that wastewater monitoring will not provide individual-level assessment but rather serves as a complementary approach to understanding community-level spread of the disease. In lines 46-54 of the Introduction, we discuss several benefits of wastewater monitoring compared to clinical testing. We added to the second paragraph of the Future Perspectives (lines 388-393) to highlight how officials can use information about the monitored sewered population to increase the utility of wastewater monitoring.

“A better understanding of the characteristics of populations included in wastewater monitoring will also help officials use wastewater data effectively and adapt sampling strategies to address ongoing public health needs. Insights into the demographics and vulnerability of wastewater-monitored populations can enable tailored communications and interventions and equitable resource distribution. Furthermore, wastewater programs can use population information to effectively monitor additional pathogens such as influenza, respiratory syncytial virus, and monkeypox virus.”

• Point 3: The results from the study can be briefly justified based on the authors’ experience or understanding

a. For example-

Line 260-263 - Educational attainment was lower in the sewered population than the unsewered population in seven counties (all but Forsyth County and Pitt County), ranging from a -7.0 to -0.2 pp difference. This difference was also statistically significant for Durham and Chatham counties (Table S4).

What is the reason for this? How is this expression related to the aim of the study? The authors can explain the results after stating them with the reason as per their experience or understanding.

Author response: We appreciate this feedback. We added a brief explanation of the results based on our knowledge of North Carolina’s wastewater program and two new references in lines 189-195

“The observed differences may be related to how sites were enrolled in North Carolina’s wastewater monitoring program. The initial group of sites participating in the NCWMN came from a COVID-19 wastewater surveillance pilot project coordinated by universities [22], and so were located in urban centers near universities with labs that had the capacity to analyze wastewater samples. Over time, the NCWMN expanded to include sites in other areas of the state, including the rural mountainous region in Western North Carolina, and underserved communities with higher social vulnerability, low COVID-19 vaccination rates, or both [23].”

• Point 4: The discussion section explains more about the result, limitations and future work; however, a generalized discussion about integrating the findings of the study to provide a comprehensive picture and its importance in the context of existing literature needs to be added apart from just one the statement in line number 319-320 “Furthermore, given these disparities, wastewater monitoring data should be interpreted alongside other surveillance data to gain a more complete picture of the ‘true’ state of public health”

Author response: We thank the reviewer for suggesting ways to improve the manuscript’s relevance. We added to the discussion section to include how this study fits in the context of existing literature in lines 273-280 and added one reference.

“Wastewater data can be used to track disease trends in communities connected to a sewer system. However, research describing how sewered and unsewered populations differ is limited. One study utilized WWTP sewershed polygon data to examine differences in population sizes, but did not look at social vulnerability or demographic differences [24]. Another explored demographic and economic characteristics of US households connected to sewer aggregated to various Census geographies, but did not utilize sewershed boundary data [25]. The present study is the first to combine sewershed polygon data and several population-based spatial datasets to characterize sewered and unsewered populations and assess whether populations monitored by a state’s wastewater program represent broader populations.”

• Point 5: Conclusion, is a stand-alone section that should also include the limitation and future scope. In the conclusion section, the authors have precisely stated the outcomes of the study. However, the authors also need to summarize the limitations and future scope to make it a distinct section for effective readership.

Author response: Thank you for bringing this to our attention. We re-worded the Conclusion to include limitations and future perspectives in lines 406-410.

“The in-depth geospatial analyses described here provide a framework for evaluating the characteristics and representativeness of wastewater-monitored populations and can be adapted as additional geospatial data describing sewered areas and population characteristics become available. Understanding sewered population characteristics will help officials use wastewater data effectively for public health decision-making and adapt wastewater testing strategies to monitor for future pathogens.”

• Point 6: Minor corrections:

a. At some places % or pp is written, be consistent about its use throughout the manuscript

Author response: Thank you. Lines 151-154 stated that % and pp have different meanings. We used percentage point (pp) when we calculated the difference for categorical variables and we used percent (%) for calculating the relative difference in continuous variables (e.g., median household income). We also confirmed that the correct abbreviations were used for variables throughout the paper.

b. Supplementary sheet – Include the title, authors and affiliation at the top of the supplementary sheet so that the file doesn't get lost when a reader downloads it.

Author response:

Thank you. We have added the title, authors, and affiliations to the top of the supplementary sheets.

---

## [Editor Report · Decision Letter 2]

20 Sep 2024

Equity in wastewater monitoring: Differences in the demographics and social vulnerability of sewered and unsewered populations across North Carolina

PONE-D-24-16609R2

Dear Dr. Reckling,

We’re pleased to inform you that your manuscript has been judged scientifically suitable for publication and will be formally accepted for publication once it meets all outstanding technical requirements.

Kind regards,

Renjith VishnuRadhan, PhD

Academic Editor

PLOS ONE